# Temporal dynamics of viral fitness and the adaptive immune response in HCV infection

Melanie Rose Walker[1,2†], Preston Leung[1†], Elizabeth Keoshkerian[1,2], Mehdi R Pirozyan[1,2], Andrew Lloyd[2], Fabio Luciani[2*†], Rowena A Bull[1,2†]

[1]Viral Immunology Systems Program, The Kirby Institute, Sydney, Australia; [2]School of Biomedical Sciences, Faculty of Medicine, The University of New South Wales, Sydney, Australia

---

## eLife Assessment

The authors examined the evolution of hepatitis C virus (HCV) in a cohort of 14 subjects with recent HCV infections. They showed that viral fitness declines as the virus mutates to escape the immune response and can rebound later in infection as HCV accumulates additional mutations. The study contributes to an **important** aspect of viral evolution. The combination of approaches contributes to a **convincing** study.

---

**\*For correspondence:**
luciani@unsw.edu.au

†These authors contributed equally to this work

**Competing interest:** The authors declare that no competing interests exist.

## Abstract

Numerous studies have shown that viral variants that elude the host immune response may incur a fitness expense, diminishing the survival of the viral strain within the host, and the capacity of the variant to survive future transmission events. This definition can be divided into intrinsic fitness—fitness without immune pressure—and effective fitness, which includes immune influence. Co-occurring mutations outside immune-targeted epitope regions may also affect variant survival (epistasis). Analysis of viral fitness and epistasis over the non-structural protein regions is lacking for hepatitis C virus (HCV). Using a rare cohort of subjects recently infected with HCV, we build on prior work by integrating mathematical modeling and experimental data to examine the interplay between transmitted/founder (T/F) viruses, immune responses, fitness, and co-occurring mutations. We show that viral fitness declines during the first 90 days post-infection (DPI), associated with the magnitude of CD8 +T cell responses and early diversification. Fitness then rebounds in a complex pattern marked by co-occurring mutations. Finally, we demonstrate that an early, strong CD8 +T cell response in the absence of neutralizing antibodies (nAbs) exerts strong selective pressure, allowing escape and chronic infection. These insights support HCV vaccine strategies that elicit broad T and B cell immunity.

## Introduction

Hepatitis C virus (HCV) is a major cause of chronic liver disease globally (*Lanini et al., 2016*). Following acute infection, approximately 75% of people fail to clear the virus, resulting in chronic hepatitis with progressive fibrosis and ultimately cirrhosis, liver failure, and an increased risk of hepatocellular carcinoma (*Seeff, 2002*; *Lavanchy, 2009*; *Micallef et al., 2006*; *Santantonio et al., 2008*). Despite the arrival of direct-acting antiviral agents (DAA) with remarkable efficacy, the unresolved financial and health service challenges in ensuring universal DAA access to those infected globally, and likelihood of reinfection in high-risk populations, necessitate a prophylactic vaccine as an essential component for the WHO goal of global elimination of HCV infection as a public health threat (*Fuerst et al., 2017*).

Due to a high mutation rate during replication, the HCV genome is highly diverse, being classified into eight major genotypes (genotypes 1–8), and 86 subtypes (labeled alphabetically [a, b, c, etc.]) (*Smith et al., 2014*; *Hedskog et al., 2019*). HCV also exists within each infected host as a diverse, rapidly evolving population termed a quasispecies. Nevertheless, transmission of HCV is associated with a strong genetic bottleneck with as few as 1–3 transmitted/founder (T/F) viruses commonly establishing infection in the new host, despite hundreds of individual variants found in the source (*Li et al., 2016*; *Bull et al., 2011*). This is followed by a second genetic bottleneck at ~100 days post-infection (DPI), where T/F viruses become undetectable. At this stage, either an existing variant that was occurring in low frequency outside detection range or an existing variant with novel mutations generated following immune selection is observed in those who progress to chronic infection. These variants carry mutations within epitopes targeted by B cell and CD8 +T cells (*Bull et al., 2011*). Indeed, clearance of HCV has been associated with an early onset of T/F-specific neutralizing antibodies (nAbs) (*Walker et al., 2019*; *Dowd et al., 2009*; *Lavillette et al., 2005*; *Osburn et al., 2014*), while early and strong CD8 +T cell responses against the T/F virus have been associated with the generation of immune escape mutations found within MHC-I restricted epitopes which lead to the progression of chronic infection (*Bull et al., 2015*; *Cai et al., 2022*).

Numerous virological studies have shown that viral variants that elude the host immune response might incur a fitness expense, diminishing the survival of the viral strain and reducing the capacity of the variant to survive future transmission events as the variants mutate away from the T/F virus (*Sanjuán et al., 2004*; *Venner et al., 2016*). This generic definition of fitness can be further divided into intrinsic fitness (also referred to as replicative fitness), where the fitness of sequence composition of the variant is estimated without the influence of host immune pressure. On the other hand, effective fitness (from here on referred to as viral fitness) considers fundamental intrinsic fitness with host immune pressure acting as a selective force to direct mutational landscape (*Hart and Ferguson, 2015*), which subsequently influences future transmission events as it dictates which subvariants remain in the quasispecies. In HCV, the structural envelope proteins, E1 and E2, which are predominantly targeted by B cell responses including neutralizing antibodies (nAbs), have been found to collectively mediate viral fitness (*Zhang et al., 2023*). This interdependence suggests that mutations in the E1 protein could potentially compensate for fitness deficits, thereby facilitating the virus' ability to evade antibodies that specifically target the E2 protein (*Zhang et al., 2023*). Furthermore, comparative analyses of E2 fitness in genotype 1 a and genotype 1b sequences found that subtype 1b viruses have a higher probability to evade immune responses (*Quadeer et al., 2019*; *Zhang et al., 2022*). By contrast, a comprehensive analysis of the non-structural protein regions (NS1, NS2, NS3, NS4A, NS4B, NS5A, NS5B) in mediating viral fitness is lacking. For the scope of this study, a fitness model developed for HIV (*Ferguson et al., 2013*; *Barton et al., 2016*) and modified for HCV (*Hart and Ferguson, 2015*) was used to estimate the fitness landscape of HCV subtypes. Barton et al.'s approach to understand HIV mutational landscape resulting in immune escape had two fundamental points: (1) replicative fitness depends on the virus sequence and the requirement to consider the effect of co-occurring mutations, and (2) evolutionary dynamics (e.g. host immune pressure) (*Barton et al., 2016*). Together they pave the way to predict the mutational space in which viral strains can change given the unique immune pressure exerted by individuals infected with HIV. This model fits well with the pathology of HCV infection. For instance, HIV and HCV are both RNA viruses with rapid rates of mutation. Additionally, like HIV, chronic infection is an outcome for HCV-infected individuals; however, unlike HIV, there is a 25% probability that individuals infected with HCV will naturally clear the virus. Previously published studies (*Keele et al., 2008*) have shown that HIV also goes through a genetic bottleneck which results in the T/F virus losing dominance and being replaced by a chronic subtype, identified by the immune escape mutations. The concepts in Barton's model and its functionality to assess fitness based on the complex interaction between viral sequence composition and host immune response are also applicable to early HCV infection. For generating the viral fitness models, sequences in clinical databases were used to represent the circulating viral variants with the assumption that commonly observed viral variants denoted higher fitness. This model was then used to infer the initial fitness landscape by providing the T/F virus and producing a fitness value. The fitness of the subsequent variants was then described as the relative fitness in comparison to the T/F virus, over the course of an infection.

Mutations that can incur a fitness expense can typically be alleviated or compensated by other mutations, making certain combinations of mutations beneficial (*Campo et al., 2008*). These co-occurring

**Table 1.** Subject characteristics and time point analysis.

| Subject ID* | Age | Sex | Disease outcome | GT† | HLA-A | | HLA-B | | First sampling point (DPI‡) | Time to clearance | Initial viral load | No. samples sequenced§ | No. T/F¶ viruses |
|---|---|---|---|---|---|---|---|---|---|---|---|---|---|
| 300023 | 22 | M | Chronic | 1 a | 02:01 | 02:01 | 44:02:00 | 57:01:00 | 36 | - | 19,234,348 | 6 | 2 |
| 300240 | 21 | M | Chronic | 3 a | 02:01 | 02:01 | 15:01 | 57:01:00 | 44 | - | 54,887 | 5 | 1 |
| 300256 | 31 | M | Chronic | 1 a | 03:01 | 24:02:00 | 07:02 | 35:01:00 | 44 | - | 34,149,824 | 5 | 1 |
| HOKD0485FX | 26 | F | Chronic | 1b | 01:01 | 30:01:00 | 08:01 | 13:02 | 30 | - | 733,849 | 8 | 1 |
| THDS1086MX | 25 | M | Chronic | 1 a | 02:01 | 32:01:00 | 14:02 | 27:05:00 | 16 | - | 235,662 | 6 | 1 |
| THGS0684MX | 28 | M | Chronic | 1 a | 02:01 | 32:01:00 | 27:02:00 | 40:01:00 | 2 | - | 140,200 | 5 | 1 |
| 300360 | 29 | M | Clearer | 3 a | 68:02:00 | 32:01:00 | 14:02 | 40:01:00 | 30 | 178 | 5,648,631 | 3 | 1 |
| 300277 | 25 | M | Clearer | 3 a | 02:01 | 11:01 | 44:02:00 | 44:02:00 | 39 | 69 | 5,482,503 | 3 | 1 |
| MCRL0786FX | 25 | F | Clearer | 1 a | 01:01 | 29:02:00 | 44:02:00 | 44:02:00 | 80 | 115 | 1846 | 1 | 1 |
| 400087 | 32 | F | Clearer | 1b | 24:02:00 | 30:04:00 | 14:02 | 15:06 | 45 | 139 | 13,118,082 | 2 | 1 |
| 300364 | 29 | M | Clearer | 1 a | 01:01 | 03:01 | 07:02 | 57:01:00 | 337 | 352 | 1932 | 1 | 1 |
| 300231 | 22 | M | Clearer | 3 a | 01:01 | 01:01 | 07:02 | 57:01:00 | 6 | 148 | 2,242,163 | 1 | 1 |
| 300089 | 26 | M | Clearer | 1b | 01:01 | 30:01:00 | 07:02 | 57:01:00 | 181 | 357 | 70,737 | 1 | 1 |
| 300164 | 22 | F | Clearer | 3 a | 24:02:00 | 32:01:00 | 07:02 | 40:01:00 | 71 | 204 | 684,028 | 1 | 1 |

*Identification.
†Genotype.
‡Days post infection.
§Next generation sequencing.
¶Transmitted/Founder.

mutations can arise outside of the epitope regions and facilitate their compensatory effects, thus increasing the likelihood of survival of the variant (*Oniangue-Ndza et al., 2011*). This phenomenon is called positive epistasis. However, epistasis may also be negative if a combination of mutations provides a smaller fitness gain than anticipated based on the additive effects of the individual mutations. Nevertheless, epistatic interactions in HCV have not been studied over acute and chronic phase sequences to infer HCV evolution over the course of the infection. Understanding viral fitness and epistatic interactions in HCV is critical in anticipating future substitutions that are antigenically significant, thus aiding in not only mapping the potential landscape of evolution, but also an improved strategic approach for the selection of potent vaccine strains.

Previously, we longitudinally deep sequenced HCV in 14 individuals who were recently infected with HCV to identify the T/F virus and the evolving quasispecies during the infection (*Bull et al., 2015*; *Cai et al., 2022*). We identified Human Leucocyte Antigen class I (HLA-I) epitopes from the T/F viruses and the mutated variants. Experimental testing using IFN-γ ELISpot revealed that a strong CD8 +T cell response was linked to rapid immune evasion (*Bull et al., 2015*; *Cai et al., 2022*). Here, we build upon our prior investigations by integrating mathematical models and experimental data to examine the interplay between evolving T/F viruses, the adaptive immune response, viral fitness, and co-occurring mutations.

## Results

### Prediction, generation, and validation of HLA class I-restricted epitopes

A total of 14 subjects with primary HCV infection followed from pre-seroconversion until infection outcome were included in this study (*Table 1*). The first available viremic sample was collected at a median of 68 days post-infection (DPI) (range 2–337 DPI). Of the 14 subjects included in this study, 8 subjects naturally cleared the infection (termed here clearers) and 6 subjects progressed to chronic

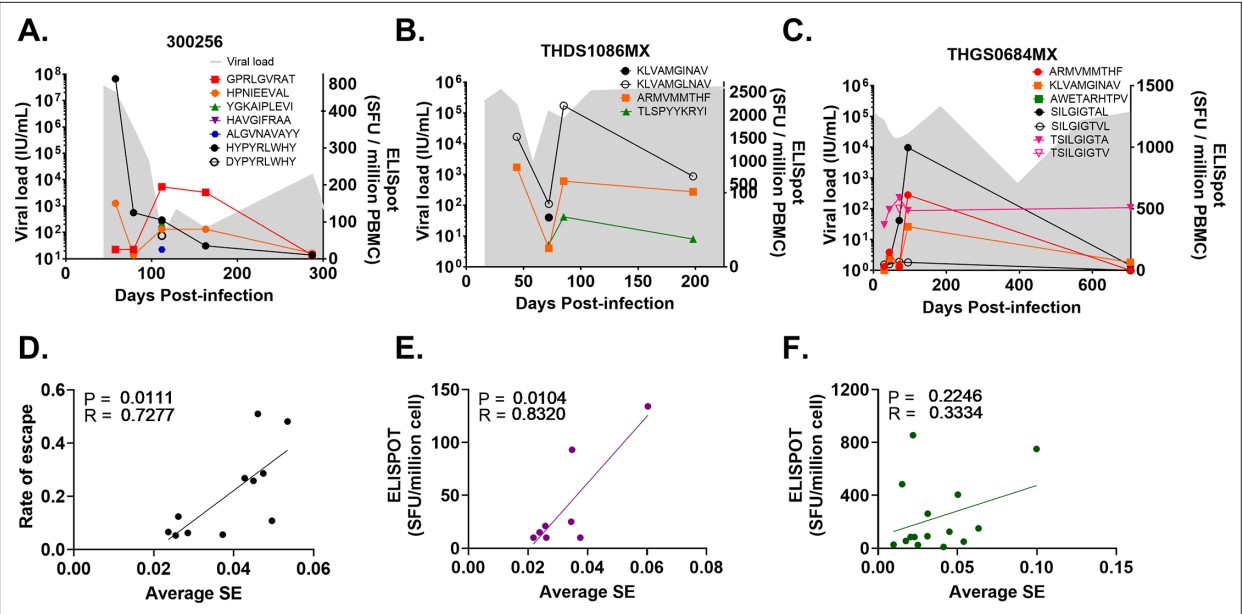

**Figure 1.** IFN-γ response and the relationship between viral diversity and the rate of immune escape in subjects chronically infected with HCV. IFN-γ ELISpot values (SFU/million, right y-axis) and viral load (IU/mL, left y-axis) measured in subjects (**A**) 300256, (**B**) THDS1086MX and (**C**). THGS0684MX for epitope-specific CD8 +T cell responses (see key). Subjects HOKD0485FX, 300023 and 300240 have been shown previously (*Bull et al., 2011*; *Cai et al., 2022*). Escape variants are shown with a clear symbol of the original epitope found in the transmitted/founder (see key). IFN-γ ELISpot values were generated from multiple biological samples. Peptides were pooled and tested in technical duplicate; positive responses were confirmed by testing the individual peptide in a follow-up IFN-γ ELISpot assay. (**D**) Plot of average Shannon entropy (SE) against the rate of escape for each epitope in each protein region per subject. Plots of average SE against average IFN-γ ELISPOT response at >90 DPI (purple) (**E**) and <90 DPI (green) (**F**) are also shown. p-Values (P) and Pearson's correlation coefficient (R) are shown in the top left corner of each panel.

The online version of this article includes the following source data and figure supplement(s) for figure 1:

**Source data 1.** Source data of IFN-γ response, viral diversity and the rate of immune escape in subjects chronically infected with HCV.

**Figure supplement 1.** Magnitude of CD8 +T cell responses associated with escape.

infection (termed here chronic progressors). The median time to natural clearance was 163 DPI (range 69–357). Next-generation sequencing (NGS) data were available for eight viremic time points from these subjects to study HCV RNA populations in depth. The T/F viruses were estimated from the distribution of variants at the earliest sampling time point as previously described (*Bull et al., 2011*; *Walker et al., 2019*; *Bull et al., 2015*). Thirteen of the 14 subjects had a single T/F virus, with only subject 300023 having two T/F viruses identified, as previously published (*Bull et al., 2011*; *Bull et al., 2015*). T/F genomes were utilized to predict HLA-I epitopes from the T/F viruses and the mutated variants (*Cai et al., 2022*). These epitopes were tested and validated for IFN-γ ELISPOT (*Figure 1—figure supplement 1*; *Cai et al., 2022*).

To examine the role of CD8 +T cell responses in driving viral evolution, longitudinal deep sequenced data was analyzed for fixation events as previously described (*Cai et al., 2022*). For those subjects who ultimately spontaneously cleared HCV infection, there were no non-synonymous fixation mutations (defined as having frequency of occurrence greater than or equal to 70% in the sequencing data) occurring within epitope regions (*Supplementary file 1*). Therefore, subjects who cleared infection were not included for further analysis of viral fitness and immune escape throughout this manuscript. From the six subjects who developed chronic infection, non-synonymous fixation mutations occurring within epitope regions were observed and peptide epitopes were selected (*Supplementary file 2*). A total of 494 out of 10945 (4.51%) potential epitopes across the six chronic subjects (45–100 epitopes per subject) were selected and tested for IFN-γ ELISPOT (*Supplementary file 2*, *Figure 1—figure supplement 1*). Of those selected for testing, 30 (6.68%) epitopes induced positive responses in the IFN-γ ELISPOT, and of these, 14 epitopes showed evidence of escape (as indicated by reduced recognition in the IFN-γ ELISPOT assay; *Figure 1*, *Supplementary file 2*, *Figure 1—figure supplement 1*).

The location of these 14 epitopes associated with escape was mostly within the non-structural regions of the HCV genome, with 7 (50%) epitopes identified in the NS3 region, 3 (21.4%) in NS5B, and 2 (14.3%) in NS2. One (7%) epitope was identified in the Core region and one (7%) in E2, which were both found in subject 300256 (*Table 2*). Single fixation events were observed in the majority of these epitopes, with the exception of two epitopes (in subjects HOKD0485FX and 300256) where two fixation events occurred (*Table 2*).

## CD8+ T-cell responses contribute to the diversification of the viral population

In these same subjects, we previously found that high magnitude of IFN-γ responses were associated with rapid viral immune escape (*Cai et al., 2022*). Additionally, the interaction between immuno-dominance, entropy, and escape rate in acute HIV infection has been described, where immuno-dominance during acute infection was the most significant factor influencing CD8 +T cell pressure, with higher immunodominance linked to faster escape (*Liu et al., 2013*). In contrast, lower epitope entropy slowed escape, and together, immunodominance and entropy explained half of the variability in escape timing (*Liu et al., 2013*). To expand on these findings and determine whether viral diversity correlated with immune escape, Shannon entropy (SE) was calculated (*Bull et al., 2011*) across each protein region at the preceding time points to those identified to have fixation in T-cell targeted epitopes, and tested (*Table 2*) against the rate of escape ($\epsilon$) estimated from the distribution of viral mutations in the NGS data. A significant positive correlation was found between SE and the rate of escape (Pearson's correlation = 0.73, p-value = 0.01), indicating the higher diversity was associated with the occurrence of immune escape (*Figure 1D*).

To determine whether CD8 +T cell pressure was driving genomic diversification before and after the genetic bottleneck, the average IFN-γ ELISPOT response within each subject was calculated across multiple epitopes in the same protein region at each time point. This data was then correlated with global SE at two periods, <90 DPI and >90 DPI. At >90 DPI a significant positive correlation was found (Pearson's correlation = 0.83, p-value = 0.01, *Figure 1E*), indicating that a stronger IFN-γ ELISPOT response was associated with higher diversity following the second genetic bottleneck. Interestingly, this relationship was not evident when examining time points at <90 DPI (Pearson's correlation = 0.33, p-value = 0.23, *Figure 1F*). When considering both pre- and post-90 DPI periods, diversity appeared to be influenced by distinct selective pressures where the CD8 +T cell response plays a significant role in shaping viral population diversity, along with the emergence of epitope escape variants at >90 DPI.

## Viral fitness decreases during the first 90 DPI and is associated with the magnitude of CD8+ T-cell responses and the initial level of diversification

To understand how CD8 +T cell responses influence the evolution of viral fitness in the acute phase of infection and how the viral population adapts to the host, a quantitative analysis of viral fitness in viral haplotypes on longitudinal samples was performed. We utilized a fitness model initially developed for HIV (*Barton et al., 2016*) and adapted for HCV to estimate the fitness landscape of various HCV subtypes [24]. This model was then applied to infer the initial fitness landscape by providing the T/F virus, which generated a fitness value. The fitness of subsequent variants was calculated as relative fitness compared to the T/F virus over the course of the infection. Specifically, fitness of the viral population was estimated for the six chronic progressors (*Table 2*) in regions NS2, NS3, and NS5B and for each epitope, the population average relative fitness of the viral population over the course of infection was measured (*Tables 3 and 4*, *Supplementary files 3–6*).

In general, in the six subjects who progressed to chronic infection, there was a decrease in average relative fitness in the period of <90 DPI with respect to the T/F virus (*Figure 2*). This was with the exception of subject 300023 which can be explained by the presence of two T/F viruses (*Figure 2A*) where the observed mutations responsible for the increase in viral fitness were present in the genome of the second T/F virus detected at 44DPI carrying substitutions H2750Q and T2917A (*Supplementary file 3*, *Figure 2—figure supplement 1*, *Figure 2A*; *Bull et al., 2011*; *Walker et al., 2019*).

To understand the observed reduction in viral fitness within the first 90 DPI, a statistical analysis was performed to examine the relationship between population fitness and immune escape ($\epsilon$), IFN-γ ELISPOT responses, and SE values (*Figure 3*). In the <90 DPI period, a significant positive correlation

**Table 2.** Epitopes and escape rate of epitopes from chronic progressors with fixation events where positive IFN-$\gamma$ ELISPOT response was detected on the T/F virus.

| Subject ID | GT | Epitope (MT)*,† | Epitope (WT)† | Start aa‡ | End aa | Region | ELISPOT‡ | DPI§ | Matching Subject Allele | Predicted Rank (WT) | Rank Change | Epsilon Estimate¶ | p-Value | Fitness Estimated |
|---|---|---|---|---|---|---|---|---|---|---|---|---|---|---|
| 300023 | 1a | RAEAHLHAW | RAEAQLHAW | 852 | 860 | NS2 | 91 | 60 | HLA-B*57:01 | 0.3 | 0 | 0.0625 | 0.0001 | Yes |
| | | NSKRTPMGF | KSKRTPMGF | 2629 | 2637 | NS5B | 484 | 60 | HLA-B*57:01 | 0.45 | -2.2 | 0.1243 | 0.0036 | Yes |
| | | RAQALPPSW | RAQAPPPSW | 1602 | 1610 | NS3 | 55 | 44 | HLA-B*57:01 | 0.2 | 0 | 0.1025 | 0.0236 | Yes |
| 300240 | 3a | RLGPVQNEI | RLGPVQNEV | 1633 | 1641 | NS3 | 300 | 71 | HLA-A*02:01 | 1.6 | -2.1 | 0.0303 | 0.0001 | Yes |
| | | DYPYRLWHY | HYPYRLWHY | 610 | 618 | E2 | 750 | 58 | HLA-A*24:02 | 1.45 | -0.5 | 0.4813 | 0.0242 | N/A |
| | | GPKMGVRAT | GPRLGVRAT | 41 | 49 | Core | 25 | 58 | HLA-B*07:02 | 0.4 | -1.4 | 0.0532 | 0.0413 | N/A |
| 300256 | 1a | HPSIEEVAL | HPNIEEVAL | 1359 | 1367 | NS3 | 150 | 58 | HLA-B*35:01 | 0.5 | 0 | 0.0565 | 0.0001 | Yes |
| HOKD0485FX | 1b | HSRRKCDEL | HSKKKCDEL | 1395 | 1403 | NS3 | 55 | 79 | HLA-B*08:01 | 3.3 | 0.5 | 0.2685 | 0.972 | Yes |
| | | KLVAMGLNAV | KLVAMGINAV | 1406 | 1415 | NS3 | 85 | 72 | HLA-A*02:01 | 0.75 | 0.5 | 0.5098 | 0.0005 | Yes |
| | | TLSPYYKRHI | TLSPYYKRYI | 830 | 839 | NS2 | 30 | 72 | HLA-A*02:01 | 3.35 | -6.05 | 0.0689 | 0.0003 | Yes |
| THDS1086MX | 1a | VRMVMMTHF | ARMVMMTHF | 2841 | 2849 | NS5B | 25 | 72 | HLA-B*27:05 | 0.2 | -0.1 | 0.2582 | 0.0034 | Yes |
| | | TSILGIGTV | TSILGIGTA | 1324 | 1332 | NS3 | 230 | 58 | HLA-A*02:01 | 32 | 17 | 0.2861 | 0.0145 | Yes |
| | | SILGIGTVL | SILGIGTAL | 1325 | 1333 | NS3 | 405 | 58 | HLA-A*02:01 | 5.6 | -1.2 | 0.2861 | 0.0145 | Yes |
| THGS0684MX | 1a | AWETARYTPV | AWETARHTPV | 2816 | 2825 | NS5B | 50 | 58 | HLA-A*02:01 | 36.5 | 3.5 | 0.1082 | 0.0484 | Yes |

*Final most dominant escape variant is shown (i.e. frequency of occurrence >70%).

†Red shows the positions which undergo amino acid change.

‡ IFN-γ ELISPOT test of autologous PBMC measured in SFU per million PBMC against wild type epitope at earliest sample time point with a positive assay (shown in DPI column). WT - Wild Type and MT - Mutant Mutant epitopes are sourced from the final available deep sequenced data point.

§Days post infection.

¶The epsilon estimate is the rate of escape given in per day.

**Table 3.** Co-occurring mutations, epitope escape mutants, and the associated frequency of occurrence and relative fitness of the NS3 region of subject THDS1086MX.

| Time | Viral load (IU/ml) | Frequency | Relative fitness | $_{1406}$KLVAMGLNAV$_{1415}$ mutations | Co-occurring mutations[†] | | |
|------|--------------------|-----------|------------------|----------------------------------------|---------------------------|---|---|
| | | 61.70% | 1.000 | | | | |
| | | 18.10% | 1.000 | | | | |
| | | 6.00% | 1.000 | | | | |
| | | 4.80% | 0.364 | V1408A | | | |
| | | 4.60% | 1.000 | | | | |
| | | 1.70% | 1.000 | | | | |
| | | 1.60% | 0.364 | V1408A | | | |
| 16DPI | 235662 | 1.40% | 1.000 | | | | |
| | | 48.50% | 0.453 | | H1115T | | |
| | | 24.70% | 0.453 | | H1115T | | |
| | | 6.20% | 0.453 | | H1115T | | |
| | | 6.20% | 0.453 | | H1115T | | |
| | | 3.70% | 0.084 | | H1115T | C1318Y | |
| | | 3.60% | 0.453 | | H1115T | | |
| | | 3.30% | 0.453 | | H1115T | | |
| | | 2.10% | 0.084 | | H1115T | | |
| 72DPI | 176550 | 1.80% | 0.069 | | H1115T | C1318Y | V1458A |
| | | 35.80% | 0.223 | I1412V | H1115S | R1118H | |
| | | 19.10% | 0.384 | I1412L | H1115S | | |
| | | 8.60% | 0.910 | I1412L | V1112A | H1115S | |
| | | 7.70% | 0.360 | I1412L | V1112I | H1115S | |
| | | 6.30% | 0.061 | I1412L | H1115G | | |
| | | 5.70% | 0.247 | I1412L | H1115T | | |
| | | 3.70% | 0.657 | I1412L | V1112T | H1115S | |
| | | 2.60% | 0.145 | I1412L | V1112A | H1115T | |
| | | 2.30% | 0.062 | I1412L | V1112I | H1115G | |
| | | 2.20% | 0.229 | I1412L | V1112I | H1115T | |
| | | 2.20% | 0.573 | I1412L | V1112A | H1115T | |
| | | 2.10% | 0.624 | I1412V | H1115S | | |
| 109DPI | 108737 | 1.80% | 0.309 | I1412L | H1115A | | |

*Table 3 continued on next page*

*Table 3 continued*

| Time | Viral load (IU/ml) | Frequency | Relative fitness | $_{1406}$KLVAMGLNAV$_{1415}$ mutations | Co-occurring mutations[†] | | |
|---|---|---|---|---|---|---|---|
| | | 73.60% | 0.360 | I1412L | V1112I | H1115S | |
| | | **12.60%** | **2.149** | **I1412L*** | **V1112I** | **H1115S** | **A1593V** |
| | | **4.70%** | **1.683** | **A1409T** | **V1112P** | **H1115S** | |
| | | 2.90% | 0.360 | I1412L | V1112I | H1115S | |
| | | 1.80% | 0.360 | I1412L | V1112I | H1115S | |
| | | **1.70%** | **1.683** | **A1409T** | **V1112P** | **H1115S** | |
| | | 1.60% | 0.066 | I1412L | V1112I | H1115S | M1268I |
| 198DPI | 681389 | 1.20% | 0.360 | I1412L | V1112I | H1115S | |

*Bold - indicate escape variants with fitness estimate ≥1.

[†]Only non-synonymous mutations are shown.

(Pearson's correlation = 0.84, p-value = 0.0004) was identified between population average relative fitness and IFN-γ ELISPOT response (**Figure 3A**). A significant negative correlation between fitness estimate and SE (Pearson's correlation = –0.48, p-value = 0.01) was also identified in the <90 DPI period (**Figure 3B**). No significant correlation was observed between ϵ and fitness estimates at <90 DPI (Pearson's correlation = 0.34, p-value = 0.32; **Figure 3C**).

Similar to the analysis performed at <90 DPI, correlation tests were performed to understand the change in fitness at >90 DPI (**Figure 3**). No significant correlations were observed between the average relative fitness estimates and IFN-γ ELISPOT (Pearson's correlation = 0.55, p-value = 0.25, **Figure 3D**), SE (Pearson's correlation = 0.26, p-value = 0.26, **Figure 3E**), and ϵ (Pearson's correlation = 0.39, p-value = 0.31, **Figure 3F**).

Overall, these results suggest that in the early phase when the T/F virus was the major variant, a fit viral population was targeted by a strong CD8 +T cell response. Mutations away from the T/F virus then reduced fitness. Insignificant correlations observed in >90 DPI also confirmed that once the

**Table 4.** Co-occurring mutations, epitope escape mutants, and the associated frequency of occurrence and relative fitness of NS3 region of subject THGS0684MX.

| Time | Viral load (IU/ml) | Frequency | Relative fitness | $_{1325}$SILGIGTAL$_{1333}$ Mutations | Co-occurring mutations* | |
|---|---|---|---|---|---|---|
| | | 87.9% | 1.000 | | | |
| | | 4.8% | 0.148 | | G1076W | |
| | | 3.2% | 1.000 | | | |
| | | 3% | 0.155 | | G1076R | |
| 2DPI | 140,200 | 1.1% | 0.022 | | G1076W | Y1521C |
| | | 74.2% | 1.000 | | | |
| | | 10.8% | 1.000 | | | |
| | | 8.4% | 0.322 | | | |
| | | 5.5% | 0.162 | | M1649T | |
| 58DPI | 19,932 | 1% | 0.322 | | S1289G | |
| | | **90.9%** | **1.042** | **A1332V**[†] | **P1496S** | |
| 184DPI | 221,964 | 9.1% | 0.157 | A1332V | D1316G | P1496S |

*Only non-synonymous mutations are shown.

[†]Bold - indicate escape variants with fitness estimate ≥1.

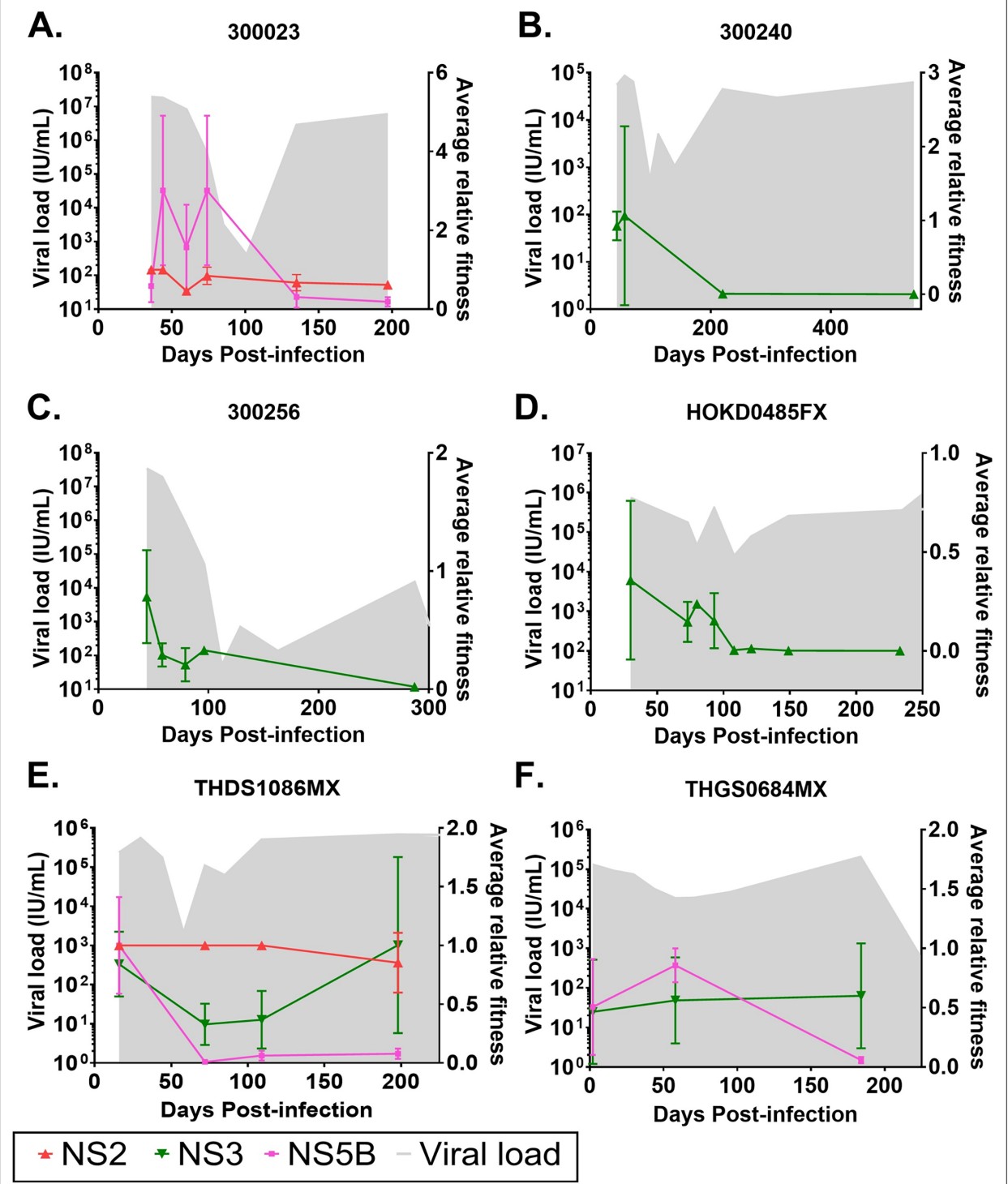

**Figure 2.** Longitudinal fitness plots of subjects chronically infected with HCV. Longitudinal fitness plots of subjects (**A**) 300023, (**B**) 300240, (**C**) 300256, (**D**) HOKD0485FX, (**E**) THDS1086MX and (**F**) THGS0684MX are shown. Gray shade indicates viral load and is measured in IU/ml on the right y-axis. Colored lines indicate population average relative fitness estimate (right y-axis) for protein regions (see key). Vertical bars indicate standard deviation of population average relative fitness.

The online version of this article includes the following source data and figure supplement(s) for figure 2:

**Source data 1.** Source data containing Days post infection, viral load and NS2, NS3, NS5B fitness estimate data.

**Figure supplement 1.** Phylogenetic analysis of subject 300023 NS5B region.

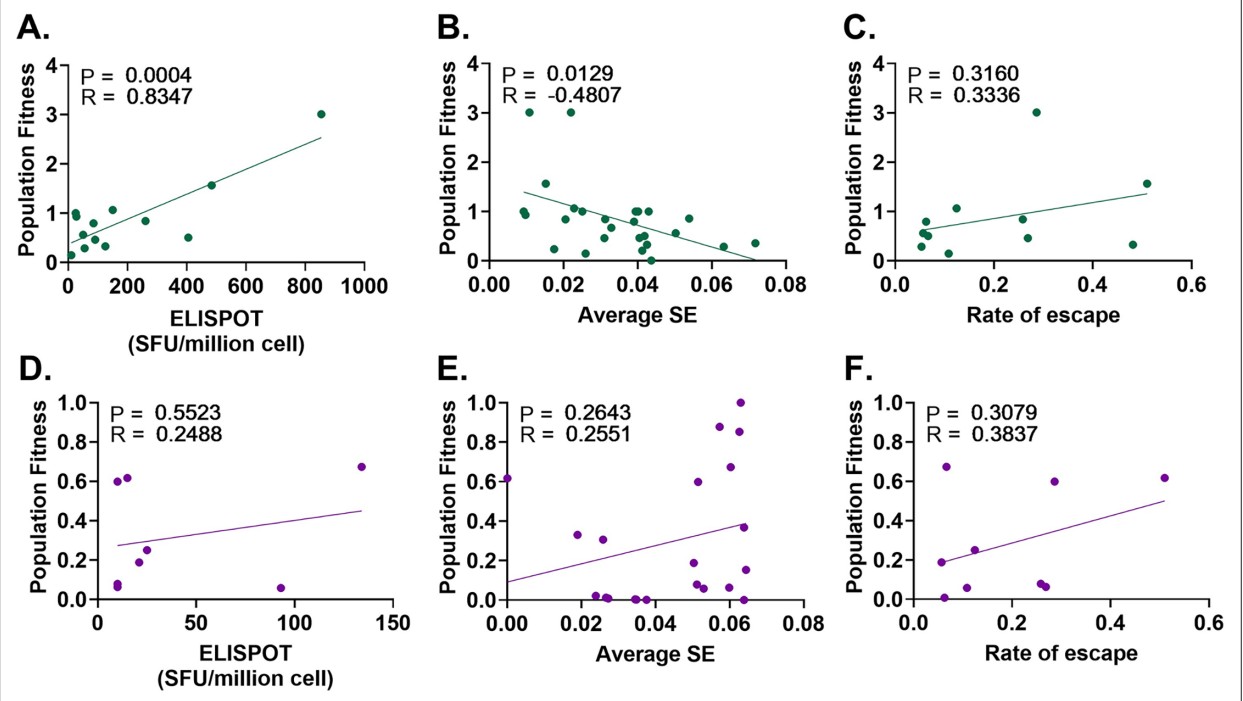

**Figure 3.** The relationship between fitness and the magnitude of the immune response (SE and IFN-γ ELISPOT) and the rate of escape at <90 DPI and >90 DPI. The relationship of population fitness against (**A**) average IFN-γ ELISPOT, (**B**) average Shannon entropy (SE), and (**C**) rate of escape at <90 DPI (green) was measured by Pearson's correlation. The relationship of population fitness against (**D**) average IFN-γ ELISPOT, (**E**) average Shannon entropy (SE), and (**F**) rate of escape at >90 DPI (purple) was also measured by Pearson's correlation. p-Values (P) and Pearson's correlation coefficient (R) are shown in the top left corner of each panel.

The online version of this article includes the following source data for figure 3:

**Source data 1.** Source data containing SE, IFN-γ ELISPOT and the rate of escape data at >90DPI and <90DPI.

chronic phase population had adapted against host immune response, immune pressure showed less impact on viral population fitness and diversity when compared to <90 DPI.

## Viral fitness rebounds with co-occurring mutations after the second genetic bottleneck at >90 DPI

After the second genetic bottleneck at >90 DPI, when viral load begins to rise, the distribution of relative fitness continued to decrease in most genomic regions analyzed for all six chronic progressors (*Figure 2*). However, THDS1086MX and THGS0684MX showed a contrasting pattern where the relative fitness measured for the NS3 region exceeded the fitness measured for the T/F virus, while the relative fitness decreased for the viral variants in comparison to the fitness of the T/F virus in the NS5B region (*Figure 2E–F*). Of note, THDS1086MX and THGS0684MX are purported to be recipients from the same donor and share an identical consensus sequence at <16 DPI (; *Walker et al., 2016*). Furthermore, the pair carries the same HLA-A alleles but different HLA-B alleles (*Table 1*). We wanted to explore the NS3 regions of THDS1086MX and THGS0684MX to further understand the specific mutations contributing to their relative fitness rebound and to elucidate the mechanisms driving viral evolution and adaptation in these individuals.

Further analysis revealed a complex pattern of evolution characterized by multiple sets of co-occurring mutations (*Figure 4*, *Table 3*). In subject THDS1086MX, at 16DPI, viral haplotypes did not carry any co-occurring mutations. However, at 72DPI, a set of three mutations (H1115T, V1458A, C1318Y) was found to be co-occurring within the same viral variant, with its relative fitness estimated to be inferior to the T/F virus (relative fitness = 0.069; *Table 3*). At 109DPI, multiple combinations of co-occurring mutations were observed. In particular, immune escape mutation 1406KLVAMG(I\L)NAV1415 was identified co-occurring with H1115S/G/T and V1112A/I/T. Variants carrying I1412L and H1115S had a relative fitness of at least one third of the T/F virus. The combination of 1406KLVAMG(I\L)

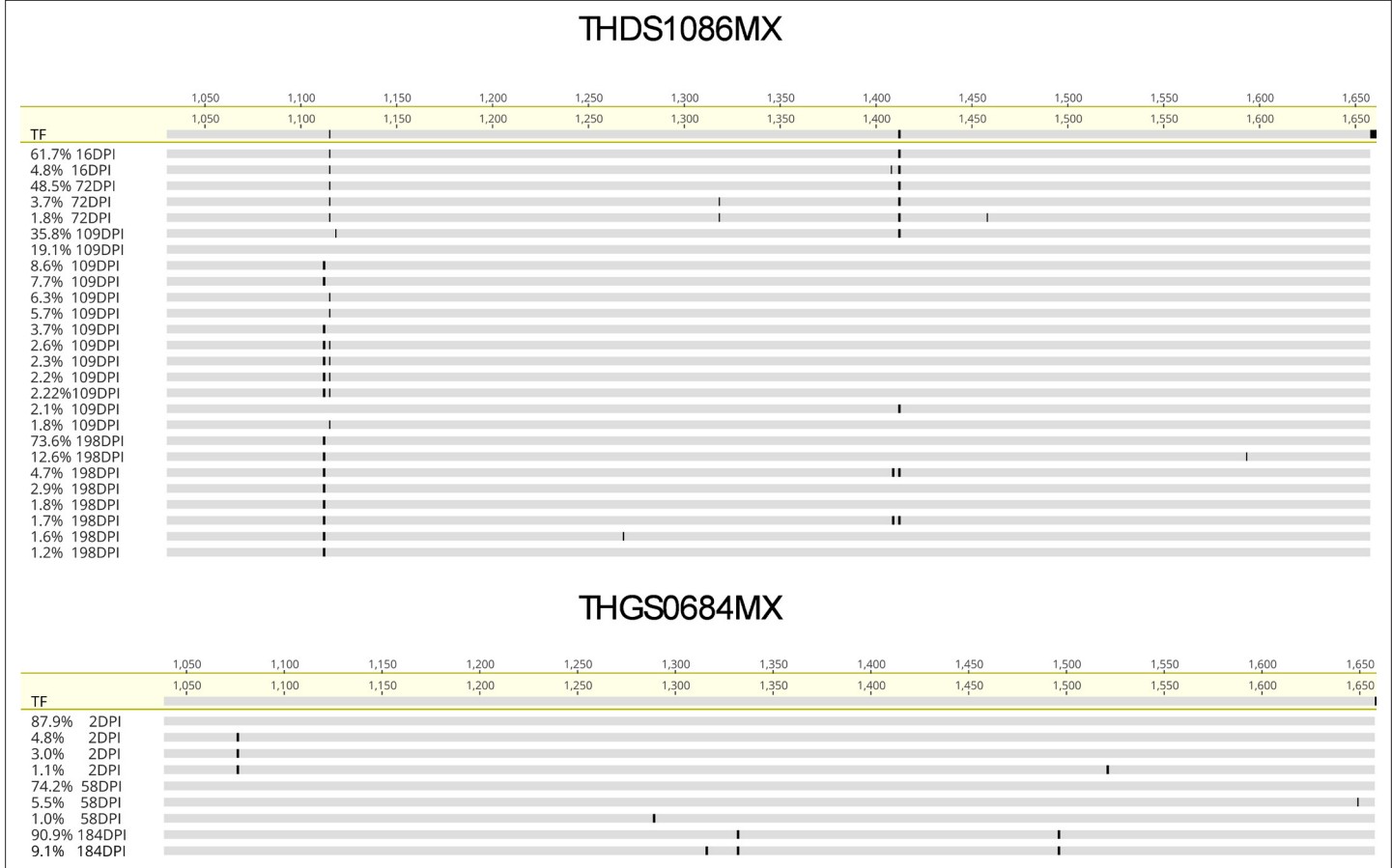

**Figure 4.** Longitudinal co-occurring mutations in the NS3 region for subjects THDS1086MX and THGS0684MX. Highlighter plots (Geneious Prime 2023) derived from longitudinal sequencing from subjects THDS1086MX (top) and THGS0684MX (bottom) indicating co-occurring mutations relative to the transmitted/founder (TF) across the NS3 region. Numbers above highlighter plots denote the genomic amino acid number for the NS3 region. Sequences are labeled by frequency of occurrence (%) and days post infection (DPI). Specific amino acid changes are shown in *Tables 3 and 4*.

NAV1415, H1115S, and V1112A reached a fitness level nearly equal to the T/F variant (relative fitness of 0.91, *Table 3*). This suggested a 90% restoration of fitness compared to the T/F virus when H1115S and V1112A were combined with the immune escape mutation. Notably, a variant with only $_{1406}$KLVAMG(I\L)NAV$_{1415}$ and H1115S exhibited positive epistasis, with a relative fitness of 0.384.

Viral variants with higher fitness than the T/F virus of subject THDS1086MX were observed at 198DPI (*Figure 4*, *Table 3*). One variant carried the epitope mutation $_{1406}$KLVAMG(I\L)NAV$_{1415}$ with V1112I, H1115S, and an additional mutation A1593V, achieving a relative fitness of 2.14 at a frequency of 12.6%. Another variant with only $_{1406}$KLVAMG(I\L)NAV$_{1415}$ V1112I and H1115S, occurring at 73.6%, showed a relative fitness of 0.360. Additionally, a new immune escape mutation, A1409L, co-occurred with V1112P and H1115S at a frequency of 4.70% and exhibited a relative fitness greater than the T/F (relative fitness = 1.68). The combination of $_{1406}$KLV(A\T)MGINAV$_{1415}$, V1112P, and H1115S did not co-occur with $_{1406}$KLVAMG(I\L)NAV$_{1415}$, V1112I, and H1115S, indicating a diversifying strategy for host adaptation. Overall, the combination of I1412L, V1112I, and H1115S emerged as the most advantageous mutation combination, enhancing virus fitness and reaching fixation in the viral population over the course of infection.

For THGS0684MX, co-occurring mutations in the NS3 region occurred in a much simpler fashion (*Table 4*), where the immune escape mutation $_{1325}$SILGIGT(A\V)L$_{1333}$ achieved a relative fitness of 1.042 when co-occurring with P1496S. The effect of the synergistic mutations was also reflected in the fact that this variant reached 90.9% frequency of occurrence in the viral population at the last sequenced time point of 184DPI (*Figure 4*, *Table 4*).

To determine if the increase in estimated fitness was associated with reversion events, namely mutations towards common circulating variants that are assumed to be a more fit HCV strain (generated as previously described *Bull et al., 2015*). A comparison of the non-synonymous mutations at the final sequence time point in both subjects THDS1086MX and THGS0684MX to the global Genbank genotype 1 a consensus sequence was performed. This revealed that V1112P and A1593V (12.6% frequency) in subject THDS1086MX were both reversions toward the worldwide consensus (*Figure 4*, *Table 3*). Similarly, for subject THGS0684MX, the mutation A1332Y (90.9% frequency) also reverted to the global consensus strain (*Figure 4*, *Table 4*).

These findings suggest rapid adaptation of chronic HCV variants post-strong CD8 +T cell responses during acute infection, supported by positive epistasis via compensatory mutations post-second genetic bottleneck. Variations between subjects with identical T/F viruses imply a unique fitness restoration strategy for each individual rather than a universal mutation pattern across subjects sharing the same T/F variant.

## In chronic progressors, nAbs emerge after CD8+ T-cells and coincide with the rebound of viral fitness

Previously, we have shown that an early nAb response is associated with clearance (*Walker et al., 2019*). In those that develop chronic infection, nAb activity is truly delayed and develops towards longitudinal variants after the T/F is cleared (*Walker et al., 2019*). Furthermore, we have shown previously (*Bull et al., 2015*; *Cai et al., 2022*) that CD8 +T cell responses impose a strong selective force on

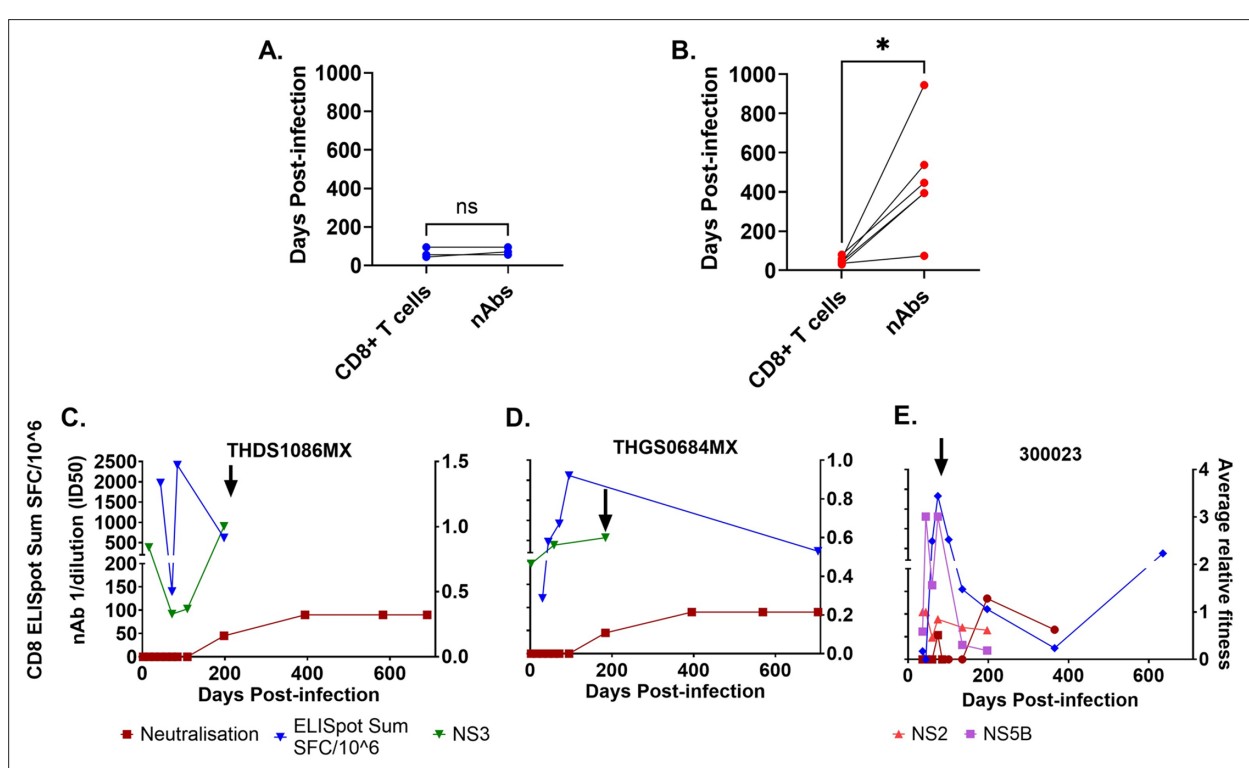

**Figure 5.** HCV neutralizing antibody (nAb), CD8 +T cell responses, and viral fitness. The timing (Days post-infection) of CD8 +T cell and nAbs is compared for clearer subjects (**A**) and chronic subjects (**B**). Statistical significance (Wilcoxon matched-pairs signed rank test) is represented by asterisks (p<0.05 (*)) and non-significance by NS. (**C-E**) show three representative subjects who developed chronic HCV infection. The blue line represents the IFN-γ ELISPOT (SFU/million). The maroon line represents HCV nAb ID50 titer with squares representing timepoints tested on autologous virus and circles representing timepoints tested on heterologous virus. Population average relative fitness estimate of regions NS5B (purple), NS3 (green), and NS2 (pink) is shown. Black arrows represent increases in average relative fitness. Neutralization results were generated from multiple biological samples with each sample assayed in technical quadruplicates across two independent experiments.

The online version of this article includes the following source data and figure supplement(s) for figure 5:

**Source data 1.** Source data containing viral load, CD8 T cell, neutralisation and fitness data.

**Figure supplement 1.** HCV neutralizing antibody (nAb) and CD8 +T cell responses and viral fitness.

the T/F virus population, thus contributing to its extinction and to the rise of the escape variants with the establishment of chronic infection. Here, a comprehensive analysis integrating nAb responses, CD8 +T cell responses, and viral fitness was performed to elucidate the dynamic interplay between the adaptive immune system and viral evolution and fitness.

All six chronic subjects included in this study had data for nAb responses (*Walker et al., 2019*), IFN-γ ELISPOT responses, and viral fitness readily available. A subset of three clearer subjects had data for nAb responses (*Walker et al., 2019*) and IFN-γ ELISPOT responses (*Figure 5*, *Figure 5—figure supplement 1*). As mentioned above, viral fitness could not be estimated for clearer subjects due to a lack of non-synonymous fixation mutations occurring within epitope regions.

The timing of the first nAb responses and the IFN-γ ELISPOT response were assessed to elucidate the dynamics of both CD8 +T cell and nAb responses and their role in clearance or chronic outcomes. In the subset of three clearer subjects (those tested for both nAb responses *Walker et al., 2019* and IFN-γ ELISPOT responses) no significant difference was found in the timing of nAb and IFN-γ ELISPOT responses, with both responses emerging in all three subjects at <100 DPI (*Figure 5A*). However, for chronic progressors, IFN-γ ELISPOT responses were first detectable at an average of 48 DPI (range 30–80 DPI) whereas nAb responses occurred significantly later, with an average onset of 465 DPI (range 74–945 DPI), indicating a notable delay compared to CD8 +T cell responses (p=0.0313, Wilcoxon matched-pairs signed rank test, *Figure 5B*).

Interestingly, in subjects THDS1086MX and THGS0684MX, we observed escape variants with higher relative fitness emerging concurrently with the onset of nAbs at 198 and 184 DPI, respectively (*Figure 5E and F*). Additionally, upon the appearance of nAbs, there was a decline in IFN-γ ELISPOT towards the selected epitopes, while nAb responses remained stable. This suggests a transition from CD8 +T cell-dominant immune responses to nAb-dominant immune responses. It's noteworthy that a similar pattern was observed in subject 300023, where an increase in viral fitness coincided with a weak but detectable nAb response towards the T/F virus at 74 DPI (*Figure 5G*). Nevertheless, nAbs were undetectable at the subsequent timepoint while the IFN-γ ELISPOT response dominated until 200 DPI when nAb responses reappeared and increased along with IFN-γ ELISPOT. This suggests that nAbs were ineffective in controlling the virus in subject 300023 during this period (<100 DPI). For the other three subjects, no consistent pattern was observed (*Figure 5—figure supplement 1*). Specifically, for subjects HOKD0485FX and 300256, nAbs emerged well after viral decline, and there was a lack of CD8 +T cell data at the positive nAb timepoints, while for subject 300240 both IFN-γ ELISPOT and nAb responses increased after 300DPI.

Together, these results show that an early and strong CD8 +T cell response in the absence of nAbs imposes a strong selective force on the T/F virus population, enabling the virus to escape and establish chronic infection.

## Discussion

Using a rare cohort of very recently infected individuals, we report the dynamics of evolution of the viral quasispecies and the role of the host cytotoxic T-cell and nAb responses during the acute phase of primary HCV infection. We show that viral fitness decreases during the first 90DPI associated with the magnitude of CD8 +T cell responses and the initial level of diversification. Thereafter, viral fitness rebounds in a complex pattern of evolution characterized by multiple sets of co-occurring mutations. Finally, within this cohort of very recently HCV-infected individuals, we show that an early and strong CD8 +T cell response in the absence of nAbs imposes a strong selective force on the T/F virus population, enabling the virus to escape and establish chronic infection. Understanding these dynamics is crucial for developing effective vaccines for HCV.

We have previously demonstrated in HCV that a genetic bottleneck occurs in the viral population at around three months post-infection, where the T/F is replaced with new viral variants that dominate infection (*Bull et al., 2011*; *Bull et al., 2015*) and this genetic bottleneck event was similarly reported in the case of early HIV infection (*Li et al., 2016*). In fact, many parallels can be drawn between HIV infections and HCV infections in the context of emerging viral species that escape T cell immune responses. For example, these new variants carry amino acid changes across the viral genome, including HLA-I restricted epitopes, highlighting that T-cell responses, in the absence of nAbs, play a significant role in driving immune escape. (*Bull et al., 2011*; *Bull et al., 2015*; *Cai et al., 2022*; *Kuntzen et al., 2007*; *Kantzanou et al., 2003*). Nevertheless, the majority of studies on HCV

CD8 +T cell responses to date focus only on the epitope and its escape variant without analyzing viral fitness or surrounding mutations which co-occur with the immune escape mutation (*Campo et al., 2014*). One major difference between HCV and HIV infection is the event where patients infected with HCV have an approximately 25% chance to naturally clear the infection as opposed to just achieving viral control in HIV infections. Here, we probed the underlying mechanism and questioned how the host immune response and HCV mutational landscape can allow the virus to escape the immune system. To understand this process, taking inspiration from HIV studies (*Barton et al., 2016*), a quantitative analysis of viral fitness relative to viral haplotypes was conducted using longitudinal samples to investigate whether a similar phenomenon was identified in HCV infections for our cohort of patients who progress to chronic infection. We observed a decrease in population average relative fitness in the period of <90 DPI with respect to the T/F virus in chronic subjects infected with HCV. The decrease in fitness correlated positively with IFN-γ ELISPOT responses and negatively with SE, indicating that CD8 +T cell responses drove the rapid emergence of immune escape variants, which initially reduced viral fitness. This is similarly reflected in HIV-infected patients where strong CD8 +T cell responses drove quicker emergence of immune escape variants, often accompanied by compensatory mutations (*Barton et al., 2016*).

While a recent HCV infection study examined mutations in the E2 region, revealing that certain co-occurring mutations led to a loss of E2 function in clearers but a gain in function for chronic progressors (*Frumento et al., 2024*), the impact of co-occurring mutations in the nonstructural regions on HCV outcomes remains largely unexplored (*Campo et al., 2008*; *Campo et al., 2014*; *Aurora et al., 2009*; *Murray et al., 2013*). Furthermore, previous studies on HCV have not integrated CD8 +T cell responses, longitudinal tracking of immune escape mutations, co-occurring mutations, and fitness estimation to analyze HCV evolutionary mechanisms (*Leung et al., 2014*). Combining these approaches provides a clearer understanding of HCV evolution by showing how co-occurring mutations impact viral survival over time. We observed that in some subjects, combinations of mutations could compensate for the negative fitness cost of immune escape, leading to a rebound in viral fitness during infection. This was not observed in all subjects and indicates that each subject's mutation strategy is unique, even when there are shared HLA alleles (such as for subjects THDS1086MX and THGS0684MX). Additionally, as described in Barton et al.'s study, it is quite possible that due to the sequence background (i.e. the T/F virus sequence composition), combined with unique immune pressure exerted by individual hosts, there exist multiple paths for immune escapes with diversifying levels of fitness (*Barton et al., 2016*).

When the analysis of nAb responses, CD8 +T cell responses, and viral fitness was performed, it was interesting to note the opposite trend with regards to the humoral response in these same subjects. While it remains unclear how T- and B-cell components of the immune system might co-operate to confer protection, the findings of the current study strongly suggest that the absence of nAbs in chronic progressors enables viruses to continue to infect new cells, replicate, and mutate. This suggests a synergistic interaction between B and T-cell responses targeting the virus, emphasizing the importance of their co-occurrence. Although several studies have alluded to this (*Walker et al., 2019*; *Bull et al., 2015*; *Cai et al., 2022*), none have directly compared longitudinal B and T-cell responses in recent clearers versus chronic progressors. In chronic subjects, increased viral fitness after the genetic bottleneck aligned with nAb onset (*Walker et al., 2019*; *Frumento et al., 2024*). Our fitness model, using the T/F virus as a baseline, enables tracking of relative viral fitness changes over time. Here, nAb appearance coincided with declining IFN-γ ELISPOT responses (*Bull et al., 2015*; *Cai et al., 2022*) towards selected epitopes that showed evidence of escape, suggesting a shift from CD8 +T cell to nAb responses.

While our findings here are promising, it should be recognized that although the bioinformatics tool (iedb_tool.py) proved useful for identifying potential epitopes, there could be epitopes that are not predicted or false positives from the output, which could lead to missing real epitopes.

In conclusion, this study provides initial insights into the evolutionary dynamics of HCV, showing that an early, robust CD8 +T cell response without nAbs strongly selects against the T/F virus, enabling it to escape and establish chronic infection. However, these findings are preliminary and not exhaustive, warranting further investigation to fully understand these dynamics. Nevertheless, the work presented here could explain the lack of association with a lower incidence of chronic HCV infection compared to placebo observed in the recent Phase I/II trials of the ChAd3-NSmut and MVA-NSmut vaccines, which

aimed to induce T cell responses by encoding HCV NS proteins (*Page et al., 2021*). While much work is still required to be able to fully understand the factors contributing to the overall dynamics of HCV infections, this work has the potential to inform the design of the next generation of vaccines.

# Methods

**Key resources table**

| Reagent type (species) or resource | Designation | Source or reference | Identifiers | Additional information |
|---|---|---|---|---|
| Cell line (*Homo sapiens*) | Lenti-X 293T Cell Line | Takara, Mountain View, CA, USA | Cat# 632180 | Cell line maintained in High Glucose Dulbecco's Modified Eagle Medium supplemented with 10% (v/v) heat-inactivated fetal bovine serum |
| Cell line (*Homo sapiens*) | Huh7.5 | Apath, New York, NY, USA | | Cell line maintained in High Glucose Dulbecco's Modified Eagle Medium supplemented with 10% (v/v) heat-inactivated fetal bovine serum |
| Transfected construct (*Photinus pyralis*) | pTG126 MLV luciferase | Prof. Francois-Loic Cosset; *Bartosch et al., 2003* | | Vector to produce HCVpp |
| Transfected construct (*murine leukemia virus*) | phCMV-5349 MLV gag/pol | Prof. Francois-Loic Cosset | | |
| Peptide, recombinant protein | CD8 T-cell (9-10mers) | Mimotopes | | |
| Commercial assay or kit | Mammalian Calphos transfection kit | Macherey-Nagel | Cat#631312 | For HCVpp transfection |
| Software, algorithm | ShoRAH | ShoRAH | RRID:SCR_005211 | |
| Software, algorithm | LoFreq | LoFreq | RRID:SCR_013054 | |
| Software, algorithm | Geneious | Geneious | RRID:SCR_010519 | |
| Software, algorithm | QuasiRecomb | QuasiRecomb | RRID:SCR_008812 | |
| Software, algorithm | GraphPad | GraphPad | RRID:SCR_000306 | |
| Software, algorithm | Immune Epitope Database and Analysis Resource (IEDB) | Immune Epitope Database and Analysis Resource (IEDB) | RRID:SCR_006604 | |
| Software, algorithm | Python | Python | RRID:SCR_008394 | |
| Software, algorithm | RStudio | RStudio | RRID:SCR_000432 | |
| Software, algorithm | ShoRAH | ShoRAH | RRID:SCR_005211 | |
| Software, algorithm | LoFreq | LoFreq | RRID:SCR_013054 | |
| Software, algorithm | Geneious | Geneious | RRID:SCR_010519 | |
| Software, algorithm | QuasiRecomb | QuasiRecomb | RRID:SCR_008812 | |

## Subjects and samples

Samples were available from two prospective cohorts of HCV-seronegative and aviremic, high-risk, individuals enrolled in the Hepatitis C Incidence and Transmission Studies (HITS) in prisons (HITS-p), or in the general community (HITS-c), as previously described (*Walker et al., 2019*; *Walker et al., 2016*; *Cunningham et al., 2017*; *White et al., 2014*). Risk behavior data and blood samples were collected

every six months to screen for HCV RNA positivity and seroconversion. Participants identified as early incident cases were enrolled into a sub-cohort called HITS-incident (HITS-i) upon detection of new onset viremia without antibodies, as described previously (*Walker et al., 2019*). These subjects were then frequently sampled for 12 weeks before being offered antiviral treatment at 24 weeks, allowing for the classification of outcomes as either natural clearance or chronic infection. The date of infection was estimated by subtracting the average HCV pre-seroconversion window period, which has been estimated at 51 days from the midpoint between last seronegative and first seropositive time points as previously described (*Bull et al., 2011*; *Walker et al., 2019*; *Page-Shafer et al., 2008*; *Glynn et al., 2005*). Molecular HLA typing was performed at the Institute of Immunology and Infectious Diseases in Perth, Australia, using second-generation sequencing of HLA-A/B/C genes as previously described (*Gaudieri et al., 2006*). 14 subjects with stored peripheral blood mononuclear cells (PBMCs) were selected from the HITS-i cohort. This group included eight subjects with acute HCV infection and six subjects who developed chronic infection.

PBMCs were isolated from peripheral blood via density-based centrifugation and resuspended in RPMI (Sigma-Aldrich, MO) supplemented with penicillin, streptomycin, l-glutamine, and fetal bovine serum at $1×10^6$ cells/mL.

## Viral sequencing and identification of T/Fs and longitudinal variants

Near full-length HCV genome amplification and sequencing datasets of the full-length viral genomes at multiple time points have been published previously (*Bull et al., 2011*; *Walker et al., 2019*; *Bull et al., 2015*; *Cai et al., 2022*; *Bull et al., 2016*; *Rodrigo et al., 2017*). Briefly, near full-length HCV genome amplification was performed using an nRT-PCR (*Bull et al., 2016*). Illumina (MiSeq Benchtop sequence, San Diego, USA) sequencing was performed on amplicons from longitudinal time points of the 14 subjects (*Walker et al., 2019*; *Cai et al., 2022*; *Bull et al., 2016*). A bioinformatics pipeline was used to clean and align reads generated from next-generation sequencing (NGS; *Bull et al., 2011*; *Walker et al., 2019*; *Bull et al., 2015*). To accurately detect single nucleotide polymorphisms (SNPs) from the aligned and cleaned sequences, analysis was performed to correct for random technical errors using the software ShoRAH, LoFreq, and Geneious (*Walker et al., 2019*; *Zagordi et al., 2011*; *Wilm et al., 2012*). Haplotypes were reconstructed from NGS data across the full genome using ShoRAH and QuasiRecomb (*Zagordi et al., 2011*; *Töpfer et al., 2013*). As described previously (*Bull et al., 2011*; *Walker et al., 2019*; *Bull et al., 2015*; *Cai et al., 2022*), to determine T/Fs, a statistical model, Poisson fitter, as well as phylogenetic analysis was applied to haplotypes from the first available viremic timepoint but prior to the first HCV antibodies (seroconversion) detection, for each subject to determine the T/F (*Salazar-Gonzalez et al., 2009*). Shannon Entropy (SE) was determined from the NGS data as previously described (*Bull et al., 2011*) with modifications to extend to individual protein regions (namely NS2, NS3, and NS5B). Briefly, SE was calculated using the frequency of occurrence of SNPs based on per codon position, this was further normalized by the length of the number of codons in the sequence which made up respective protein. An average SE value was calculated for each time point in each protein region for all subjects until the fixation event.

## CD8+ T-cell epitope selection

For each of the 14 subjects, non-synonymous fixation events (>70% frequency in the viral population) from the last time point sequenced were substituted into the /TF sequence, thus generating a modified sequence termed the fixation sequence.

Identification of MHC Class I restricted epitopes (9–10 amino acids) was performed by analysis of both HCV /TF and fixation sequences in conjunction with MHC Class I genotyping to predict epitopes and then compared with previous experimentally confirmed epitopes from the Immune Epitope Data Base (IEDB [https://www.iedb.org/]). This resulted in each prediction list containing at least 10,000 epitope predictions.

Due to the large number of epitopes and the lack of available PBMCs, a bioinformatics pipeline, iedb_tool, was developed in Python 2.7 to identify a subset of epitopes for further experimental and bioinformatics analyses. This software package is freely available and can be downloaded at: https://github.com/PrestonLeung/IEDBTool2 (copy archived at *Preston, 2025*). The predicted epitopes generated from IEDB were downloaded in plain text format and parsed into *iedb_predictionParser.py* with options *-lower 100.0* to extract relevant subject information (HLA genotypes, predicted epitope

sequence, start and end positions of the epitope sequence). The predicted data set was then used with the experimentally validated epitopes from IEDB. This step generated a categorized list of ranked epitopes for which predicted epitopes and HLA-I typing for each subject were matched with experimentally validated epitopes. An 80% homology threshold was set to facilitate small differences between the autologous epitope sequence and those that were previously reported from the IEDB database.

We prioritized epitopes that matched the HLA allele of the subject and classified these into three categories. Category 1 included epitopes for each subject from which the mutated variant was no longer predicted to be recognized, indicating a potential escape variant. Category 2 included epitopes from those that underwent escape, but the variant was still predicted to be recognized by CD8 +T cells and category 3 included epitopes that did not mutate during infection. A threshold of 10th percentile ranking or higher was used as a cutoff for epitope selection. For categories 2 and 3, we selected a maximum of 10 epitopes due to limitations in PBMC samples. An upper limit of 100 epitopes was set for each subject to account for the total PBMCs available for each time point in the ELISpot matrix testing approach (see below). If the above criteria gave a total number of epitopes less than 40, then epitopes from categories 2 and 3 were searched with a lowered threshold of 50th percentile ranking or higher to increase the sample size to at least 40. This refined list of ranked epitopes we refer to throughout the manuscript as "potential epitopes".

## IFN-γ ELISPOT validation

Potential epitopes generated above were analyzed using matrix IFN-γ ELISPOT assays as previously described (*Bull et al., 2015*; *Cai et al., 2022*). While the ELISpot assay detects responses from both CD4 and CD8 T cells, our peptide design (9-10mers) is strongly biased toward CD8 T-cell detection. We have therefore interpreted ELISpot responses primarily in terms of CD8 T-cell activity. Briefly, peptides were synthesized by Mimotopes Australia and used in autologous CD8 +T cell ELISpot assays in pools of ≤5 peptides per well in a matrix format. The peptides were pooled such that any two wells would only have one common peptide, and each peptide was tested in at least two separate wells.

As previously described by us (*Bull et al., 2015*; *Cai et al., 2022*), PBMC were added to a 96-well ELISpot plate (MAIPS; Millipore, USA) precoated with gamma interferon (IFN-γ; Mabtech, Sweden) at a concentration of 150,000 cells per well and incubated overnight with peptides (final concentration of 10 μg/ml). Plates were read and analyzed using an AID plate reader (Autoimmun Diagnostika GmbH, Germany). A peptide/well was concluded to be a positive if it demonstrated an interferon-γ (IFN-γ) response ≥25 spot-forming units per million cells (SFU/million) and a negative control count of zero. A confirmation experiment using the IFN-γ ELISpot assay was conducted for individual positive peptides using one peptide per well in a concentration of 200,000 cells per well. At least one well per study subject was allocated as a positive control (anti-CD3 antibody; Mabtech, Sweden), and three wells were used as negative controls. Background was defined as the mean plus 3 standard deviations of the number of spots counted across the three negative controls.

## Estimating the rate of CD8+ T-cell epitope escape

The estimation of the rate of CD8 +T cell epitope escape was performed as previously described (*Bull et al., 2015*; *Cai et al., 2022*). Briefly, the kinetics of CD8 +T cell escape was estimated with a simple population dynamics model (*Bull et al., 2015*; *Cai et al., 2022*; *Asquith et al., 2006*). The model predicts that the frequency of the escape variant $f(t)$ is: $f(t) = f_0/f_0 + (1 - f_0)e^{-kt}$, where k is the rate of escape and it is assumed that the escape variant is present at a low frequency beyond the range of detection during the initial phase of time (t) at zero, and its frequency is given by $f_0$. The escape variant was defined as any observed mutations away from the wild-type sequence that consequently becomes fixed in the population. Therefore, the frequency of the escape variant at individual longitudinal time points was the total frequencies of each epitope carrying the escape mutation regardless of the other amino acid positions. In some cases where the escape variant was not observed in the earlier time points, an estimate of $1/(n+1)$ replaces a frequency of 0, where n is the average coverage of the corresponding time point from the deep sequencing data. This model was used to estimate the rate of escape only in those cases with experimental evidence (ELISpot data generated in this study)

of CD8 +T-cell-driven immune escape. Estimates were performed in R usin ag non-linear least-squares approach for non-linear models (*R Development Core Team, 2012*).

## Estimating survival fitness of viral variants

The model used for estimating viral fitness has been previously described by *Hart and Ferguson, 2015*. Briefly, the original approach used HCV subtype 1 a sequences to generate the model for the NS5B protein region. To update the model for other regions (NS3 and NS2) as well as other HCV subtypes in this study, subtype 1b and subtype 3 a sequences were extracted from the Los Alamos National Laboratory HCV database. An intrinsic fitness model was first generated for each subtype for NS5B, NS3, and NS2 region of the HCV polyprotein. Then, using longitudinally sequenced data from patients chronically infected with HCV as well as clinically documented immune escape to describe high viral fitness variants, we generated estimates of the viral fitness for subjects chronically infected with HCV in our cohort. This was performed on each viral variant reconstructed by QuasiRecomb as previously described (*Hart and Ferguson, 2015*; *Ferguson et al., 2013*; *Barton et al., 2016*). Specifically, haplotype reconstruction was performed longitudinally for each subject in a region of approximately 600 amino acids surrounding the experimentally confirmed CD8 +T cell epitopes. The fitness landscape inference is obtained by estimating the parameter of a function $E\left(\vec{z}\right)$ representing the energy of the viral population and with $P\left(\vec{z}\right)$:

$$P\left(\vec{z}\right) = \frac{e^{-E\left(\vec{z}\right)}}{z},$$

$$E\left(\vec{z}\right) = \sum_{i=1}^{m} h_i\left(z_i\right) + \sum_{i=1}^{m}\sum_{j=i+1}^{m} J_{ij}\left(z_i, z_j\right)$$

where $\vec{z}$ is the peptide sequence of length $m$ such that $\vec{z} = \{z_1, z_2, \ldots z_m\}$ describes individual amino acids in $\vec{z}$. $P\left(\vec{z}\right)$ is the prevalence fitness taken under the assumption that the frequently occurring variants represent the fitter strain (i.e. the common circulating virus, in this case the original T/F virus), and this mathematically defined fitness correlates positively to the actual viral fitness $f\left(\vec{z}\right)$ (*Ferguson et al., 2013*). $E\left(\vec{z}\right)$ is the 'energy' of peptide $\vec{z}$ a pseudo measurement for fitness, and $z$ is a normalization factor described in the following equation.

$$z = \sum_{z_i=\{1,0\}} e^{-E\left(\vec{z}\right)}$$

The parameter $h_i\left(z_i\right)$ is the contribution of energy of the single amino acid at position $i$ of $\vec{z}$. $J_{ij}\left(z_i, z_j\right)$ specifies the energy contribution associated with pairwise interactions between two amino acids in $\vec{z}$ at positions $i$ and $j$. Under the model assumptions, the association between measurement of energy and fitness estimate showed a negative correlation where minimal $E\left(\vec{z}\right)$ maximizes the logarithm of the viral fitness $\log\left(f\left(\vec{z}\right)\right)$ (*Hart and Ferguson, 2015*).

A relative fitness estimate was also calculated for reconstructed haplotypes by normalizing individual fitness estimates with respect to the fitness of the haplotype corresponding to the T/F variant such that the T/F variant will have a fitness of 1.000. These values were used to represent the viral fitness of individual haplotypes to infer viral evolution dynamics with respect to the T/F virus.

This analysis was used to quantify the selection exerted by the host cytotoxic T cell response in driving the evolution of the T/F virus. The analysis also showed the role of specific co-evolving mutations and the identification of epistatic interactions that may determine the evolution of viral fitness and the overall success of the infection.

## Cell lines

Human cell lines, Lenti-XTM 293 T (Takara, Mountain View, CA, USA) and Huh7.5 (Apath, New York, NY, USA) were cultured at 37 °C in a humidified atmosphere containing 5% $CO_2$, using High Glucose Dulbecco's Modified Eagle Medium (HG-DMEM; Gibco, Thermo Fisher Scientific, Waltham, MA,

USA) supplemented with 10% (v/v) heat-inactivated fetal bovine serum (FBS; Gibco). These cells were authenticated by STR profiling and were confirmed to be mycoplasma negative.

## HCVpp production, infection, and neutralization

As previously described by us, E1E2 glycoproteins were cloned and co-transfected with MLV gag/pol and luciferase vectors in Lenti-X 293T -cells to produce HCV-pseudo-particles (pp) (*Walker et al., 2020*; *Walker et al., 2019*). Briefly, neutralization assays were performed by incubating HCVpp with heat-inactivated plasma for one hour at 37 °C. This mixture was applied directly to Huh-7.5 hepatoma cells (Apath, L.L.C, New York, NY, USA), and incubated for 72 hr after which luciferase activity was measured (*Walker et al., 2019*). Neutralization of HCVpp was calculated using the formula: % inhibition $= 1 -$ (inhibited activity)/('normal' activity)×100, after subtraction of negative control (pseudo-particle generated without glycoproteins) RLU, where normal activity is HCVpp incubated with plasma from a healthy donor. The 50% ID50 titer was calculated as the nAb concentration that caused a 50% reduction in RLU for each plasma/HCVpp combination tested in neutralization. All samples were also tested for neutralizing activity on control pseudo-particle VSV-G, to determine that nAbs were HCV E1E2 specific. All data were fitted using non-linear regression plots (GraphPad, Prism).

## Data visualization and statistical analysis

Data visualization for CD8 +T cell analyses was performed using ggplot2 package (*Wickham, 2016*) in R (*R Development Core Team, 2012*). Visualization of viral fitness estimates and longitudinal neutralization and CD8 +T cell data was performed using GraphPad Prism version 10.0 for Windows, GraphPad Software, La Jolla California USA (https://www.graphpad.com/). As were comparisons of timing of nAb and CD8 +T cell responses where Wilcoxon matched-pairs signed rank tests were performed. Sequence alignments for highlighter plots were performed on the variants generated here using Geneious Prime 2023 (https://www.geneious.com).

## Acknowledgements

We acknowledge Gregory R Hart and Prof. Andrew L Ferguson for their assistance with the fitness models.

## Additional information

### Funding

| Funder | Grant reference number | Author |
| --- | --- | --- |
| National Health and Medical Research Council | 1128416 | Fabio Luciani |
| National Health and Medical Research Council | 1041897 | Andrew Lloyd |
| National Health and Medical Research Council | 1084706 | Rowena A Bull |

The funders had no role in study design, data collection and interpretation, or the decision to submit the work for publication.

### Author contributions

Melanie Rose Walker, Preston Leung, Conceptualization, Data curation, Formal analysis, Investigation, Methodology, Writing – original draft, Writing – review and editing; Elizabeth Keoshkerian, Data curation, Formal analysis, Investigation, Methodology; Mehdi R Pirozyan, Data curation, Formal analysis, Methodology; Andrew Lloyd, Conceptualization, Supervision, Funding acquisition, Investigation, Writing – review and editing; Fabio Luciani, Conceptualization, Resources, Formal analysis, Supervision, Investigation, Methodology, Project administration, Writing – review and editing; Rowena A Bull, Conceptualization, Resources, Data curation, Formal analysis, Supervision, Funding acquisition, Investigation, Methodology, Project administration, Writing – review and editing

### Author ORCIDs
Melanie Rose Walker https://orcid.org/0000-0002-9731-9880
Fabio Luciani https://orcid.org/0000-0003-0666-6324

### Ethics
Human subjects: Human research ethics approvals were obtained from Human Research Ethics Committees of Justice Health (reference number GEN 31/05), New South Wales Department of Corrective Services (05/0884), and the University of New South Wales (05094, 08081), all located in Sydney, Australia. Written informed consent was obtained from the participants. All methods were performed in accordance with the relevant guidelines and regulations.

Reviewer #1 (Public review): https://doi.org/10.7554/eLife.102232.3.sa1
Reviewer #2 (Public review): https://doi.org/10.7554/eLife.102232.3.sa2
Author response https://doi.org/10.7554/eLife.102232.3.sa3

## Additional files

### Supplementary files
MDAR checklist

Supplementary file 1. Summary of epitope selection and (IFN-γ) ELISPOT assay responses in subjects who cleared infection.

Supplementary file 2. Summary of epitope selection and positive (IFN-γ) ELISPOT assay responses in subjects who developed chronic infection.

Supplementary file 3. Subject 300023 relative fitness estimate, co-occurring mutations and frequency of occurrence for each reconstructed haplotype.

Supplementary file 4. Subject 300240 relative fitness estimate, co-occurring mutations and frequency of occurrence for each reconstructed haplotype.

Supplementary file 5. Subject 300256 relative fitness estimate, co-occurring mutations and frequency of occurrence for each reconstructed haplotype.

Supplementary file 6. Subject HOKD0485FX relative fitness estimate, co-occurring mutations and frequency of occurrence for each reconstructed haplotype.

### Data availability
All data generated or analyzed during this study are included in the manuscript and supporting files. Source data files have been provided for Figures 1–5. The code is freely available and can be downloaded at GitHub (copy archived at *Preston, 2025*). Single-cell RNA seq data are available on accession number GSE196330. All the single-cell RNA-seq data are deposited with GSE196330.

The following previously published dataset was used:

| Author(s) | Year | Dataset title | Dataset URL | Database and Identifier |
| --- | --- | --- | --- | --- |
| Luciani F, Cai C, Samir J | 2022 | Single cell multi-omics reveals early elevated function and multiple fates within human progenitors of exhausted CD8+ T cells | https://www.ncbi.nlm.nih.gov/geo/query/acc.cgi?acc=GSE196330 | NCBI Gene Expression Omnibus, GSE196330 |

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
