## [Editor Report · eLife Assessment]

The authors examined the evolution of hepatitis C virus (HCV) in a cohort of 14 subjects with recent HCV infections. They showed that viral fitness declines as the virus mutates to escape the immune response and can rebound later in infection as HCV accumulates additional mutations. The study contributes to an **important** aspect of viral evolution. The combination of approaches contributes to a **convincing** study.

---

## [Referee Report · Reviewer #1 (Public review)]

Summary:

The authors examine CD8 T cell selective pressure in early HCV infection using. They propose that after initial CD8-T mediated loss of virus fitness, in some participants around 3 months after infection, HCV acquires compensatory mutations and improved fitness leading to virus progression.

Strengths:

Throughout the paper, the authors apply well-established approaches in studies of acute to chronic HIV infection for studies of HCV infection. This lends rigor the to the authors' work.

---

## [Referee Report · Reviewer #2 (Public review)]

Summary:

In this work, Walker and collaborators study the evolution of hepatitis C virus (HCV) in a cohort of 14 subjects with recent HCV infections. They focus in particular on the interplay between HCV and the immune system, including the accumulation of mutations in CD8+ T cell epitopes to evade immunity. Using a computational method to estimate the fitness effects of HCV mutations, they find that intrinsic viral fitness declines as the virus mutates to escape T cell responses. In long-term infections, they found that viral fitness can rebound later in infection as HCV accumulates additional mutations.

Strengths:

This work is especially interesting for several reasons. Individuals who developed chronic infections were followed over fairly long times and, in most cases, samples of the viral population were obtained frequently. At the same time, the authors also measured CD8+ T cell and antibody responses to infection. The analysis of HCV evolution focused not only on variation within particular CD8+ T cell epitopes, but also the surrounding proteins. Overall, this work is notable for integrating information about HCV sequence evolution, host immune responses, and computational metrics of fitness and sequence variation. The evidence presented by the authors supports the main conclusions of the paper described above.

Weaknesses:

After revision, this paper has no outstanding weaknesses. Points where further investigation is needed have been clearly identified.

---

## [Author Response]

The following is the authors’ response to the original reviews

**Reviewer #1 (Public review):**
Summary:The authors examine CD8 T cell selective pressure in early HCV infection using. They propose that after initial CD8-T mediated loss of virus fitness, in some participants around 3 months after infection, HCV acquires compensatory mutations and improved fitness leading to virus progression.Strengths:Throughout the paper, the authors apply well-established approaches in studies of acute to chronic HIV infection for studies of HCV infection. This lends rigor the to the authors' work.Weaknesses:(1) The Discussion could be strengthened by a direct discussion of the parallels/differences in results between HIV and HCV infections in terms of T cell selection, entropy, and fitness.

We have added a direct discussion of the parallels/differences between HIV and HCV throughout the discussion including at lines 308 – 310 and 315 -327.

Lines 308-310: “In fact, many parallels can be drawn between HIV infections and HCV infections in the context of emerging viral species that escape T cell immune responses.”

Lines: 315-327: “One major difference between HCV and HIV infection is the event where patients infected with HCV have an approximately 25% chance to naturally clear the infection as opposed to just achieving viral control in HIV infections. Here, we probed the underlying mechanism, and questioned how the host immune response and HCV mutational landscape can allow the virus to escape the immune system. To understand this process, taking inspiration from HIV studies (24), a quantitative analysis of viral fitness relative to viral haplotypes was conducted using longitudinal samples to investigate whether a similar phenomenon was identified in HCV infections for our cohort for patients who progress to chronic infection. We observed a decrease in population average relative fitness in the period of <90DPI with respect to the T/F virus in chronic subjects infected with HCV. The decrease in fitness correlated positively with IFN-γ ELISPOT responses and negatively with SE indicating that CD8+ T-cell responses drove the rapid emergence of immune escape variants, which initially reduced viral fitness. This is similarly reflected in HIV infected patients where strong CD8+ T-cell responses drove quicker emergence of immune escape variants, often accompanied by compensatory mutations (24).”

(2) In the Results, please describe the Barton model functionality and why the fitness landscape model was most applicable for studies of HCV viral diversity.

This has been added to the introduction section rather than Results as we feel that it is more appropriate to show why it is most applicable to HCV viral diversity in the background section of the manuscript. We write at lines 77-90:

“Barton et al.’s [23] approach to understand HIV mutational landscape resulting in immune escape had two fundamental points: (1) replicative fitness depends on the virus sequence and the requirement to consider the effect of co-occurring mutations, and (2) evolutionary dynamics (e.g. host immune pressure). Together they pave the way to predict the mutational space in which viral strains can change given the unique immune pressure exerted by individuals infected with HIV. This model fits well with the pathology of HCV infection. For instance, HIV and HCV are both RNA viruses with rapid rate of mutation. Additionally, like HIV, chronic infection is an outcome for HCV infected individuals, however, unlike HIV, there is a 25% probability that individuals infected with HCV will naturally clear the virus. Previously published studies [9] have shown that HIV also goes through a genetic bottleneck which results in the T/F virus losing dominance and replaced by a chronic subtype, identified by the immune escape mutations. The concepts in Barton’s model and its functionality to assess the fitness based on the complex interaction between viral sequence composition and host immune response is also applicable to early HCV infection.”

(3) Recognize the caveats of the HCV mapping data presented.

We have now recognized the caveats of the HCV mapping data at lines 354-256 “While our findings here are promising, it should be recognized that although the bioinformatics tool (iedb_tool.py) proved useful for identifying potential epitopes, there could be epitopes that are not predicted or false-positive from the output which could lead to missing real epitopes”

(4) The authors should provide more data or cite publications to support the authors' statement that HCV-specific CD8 T cell responses decline following infection.

We have now clarified at lines 352-353 that the decline was toward “selected epitopes that showed evidence of escape”.

Furthermore, we have cited two publications at line 352 that support our statement.

(5) Similarly, as the authors' measurements of HCV T and humoral responses were not exhaustive, the text describing the decline of T cells with the onset of humoral immunity needs caveats or more rigorous discussion with citations (Discussion lines 319-321).

We have now added a caveat in the discussion at lines 357-360 which reads

“In conclusion, this study provides initial insights into the evolutionary dynamics of HCV, showing that an early, robust CD8+ T-cell response without nAbs strongly selects against the T/F virus, enabling it to escape and establish chronic infection. However, these findings are preliminary and not exhaustive, warranting further investigation to fully understand these dynamics. “

(6) What role does antigen drive play in these data -for both T can and antibody induction?

It is possible that HLA-adapted mutations could limit CD8 T cell induction if the HLAs were matched between transmission pairs, as has been shown previously for HIV (https://doi.org/10.1371/journal.ppat.1008177) with some data for HCV (https://journals.asm.org/doi/10.1128/jvi.00912-06). However, we apologise as we are not entirely sure that this is what the reviewer is asking for in this instance.

(7) Figure 3 - are the X and Y axes wrongly labelled? The Divergent ranges of population fitness do not make sense.

Our apologies, there was an error with the plot in Figure 3 and the X and Y axis were wrongly labelled. This has now been resolved.

(8) Figure S3 - is the green line, average virus fitness?

This has now been clarified in Figure S3.

(9) Use the term antibody epitopes, not B cell epitopes.

We now use the term antibody epitopes throughout the manuscript.

**Reviewer #1 (Recommendations for the authors):**
Recommendations for improving the writing and presentation:(1) Introduction:Line 52: 'carry mutations B/T cell epitopes'. Two pointsi) These are antibody epitopes (and antibody selection) not B cell epitopes

We have corrected this sentence at line 55 which now reads: “carry mutations within epitopes targeted by B cells and CD8+ T cells”.

ii) To avoid confusion, add text that mutations were generated following selection in the donor.

For HCV, it is unclear if mutations are generated following selection or have been occurring in low frequencies outside detection range. Only when selection by host immune pressure arises do the potentially low-frequency variants become dominant. However, we do acknowledge it is potentially misleading to only mention new variants replacing the transmitted/founder population. We have modified the sentence at line 52 to read:

“At this stage either an existing variant that was occurring in low-frequency outside detection range or an existing variant with novel mutations generated following immune selection is observed in those who progress to chronic infection”

- Lines 51-56: Human studies of escape and progression are associative, not causative as implied.

Correct, evidence suggesting that escape and progression are currently associative. We have now corrected these lines to no longer suggest causation.

- Line 65: Suggest you clarify your meaning of 'easier'?

This sentence, now at line 72, has been modified to: “subtype 1b viruses have a higher probability to evade immune responses”

(2) Results:- Line 147: Barton model (ref'd in Intro) is directly referred to here but not referenced.

The reference has been added.

- The authors should cite previous HIV literature describing associations between the rate of escape and Shannon Entropy e.g. the interaction between immunodominance, entropy, and rate of escape in acute HIV infection was described in Liu et al JCI 2013 but is not cited.

We have now cited previous HIV research at line 147-151, adding Liu et al:

“Additionally, the interaction between immunodominance, entropy, and escape rate in acute HIV infection has been described, where immunodominance during acute infection was the most significant factor influencing CD8+ T cell pressure, with higher immunodominance linked to faster escape (27). In contrast, lower epitope entropy slowed escape, and together, immunodominance and entropy explained half of the variability in escape timing (27).”

- Line 319: The authors suggest that HCV-specific CD8 T cell response declines following early infection. On what are they basing this statement? The authors show their measured T cell responses decline but their approach uses selected epitopes and they are therefore unable to assess total HCV T cell response in participants (Where there is no escape, are T cell magnitudes maintained or do they still decline?). Can the authors cite other studies to support their statement?

We have now clarified that the decline was toward “selected epitopes that showed evidence of escape”. Furthermore, we also cite two studies to support our findings.

- Throughout the authors talk in terms of CD8 T cells but the ELISpot detects both CD4 and CD8 T cell responses. I suggest the authors be more explicit that their peptide design (9-10mers) is strongly biased to only the detection of CD8 T cells.

To make this clearer and more explicit we have now added to the methods section at line 433-435:

“While the ELISpot assay detects responses from both CD4 and CD8 T cells, our peptide design (9-10mers) is strongly biased toward CD8 T-cell detection. We have therefore interpreted ELISpot responses primarily in terms of CD8 T-cell activity.”

The points made in lines 307-321 could be more succinct

We have now edited the discussion (lines 307 – 321) to make the points more succinct (now lines 307-323).

Minor corrections to text, figures:- Figure 2: suggest making the Key bigger and more obvious.

We have now made the key bigger and more obvious

- Figure 3 A & D....is there an error on the X-axis...are you really reporting ELISpot data of < 1 spot/10^6? Perhaps the X and Y axes are wrongly labelled?

Our apologies, there was an error with the plot in Figure 3 and the X and Y axis were wrongly labelled. This has now been resolved.

- Figure 5: As this is PBMC, remove CD8 from the description of ELISpot.

We have now removed CD8 from the description of ELISpot in both Figure 5 and Figure S3

**Reviewer #2 (Public review):**
Summary:In this work, Walker and collaborators study the evolution of hepatitis C virus (HCV) in a cohort of 14 subjects with recent HCV infections. They focus in particular on the interplay between HCV and the immune system, including the accumulation of mutations in CD8+ T cell epitopes to evade immunity. Using a computational method to estimate the fitness effects of HCV mutations, they find that viral fitness declines as the virus mutates to escape T-cell responses. In long-term infections, they found that viral fitness can rebound later in infection as HCV accumulates additional mutations.Strengths:This work is especially interesting for several reasons. Individuals who developed chronic infections were followed over fairly long times and, in most cases, samples of the viral population were obtained frequently. At the same time, the authors also measured CD8+ T cell and antibody responses to infection. The analysis of HCV evolution focused not only on variation within particular CD8+ T cell epitopes but also on the surrounding proteins. Overall, this work is notable for integrating information about HCV sequence evolution, host immune responses, and computational metrics of fitness and sequence variation. The evidence presented by the authors supports the main conclusions of the paper described above.Weaknesses:One notable weakness of the present version of the manuscript is a lack of clarity in the description of the method of fitness estimation. In the previous studies of HIV and HCV cited by the authors, fitness models were derived by fitting the model (equation between lines 435 and 436) to viral sequence data collected from many different individuals. In the section "Estimating survival fitness of viral variants," it is not entirely clear if Walker and collaborators have used the same approach (i.e., fitting the model to viral sequences from many individuals), or whether they have used the sequence data from each individual to produce models that are specific to each subject. If it is the former, then the authors should describe where these sequences were obtained and the statistics of the data.If the fitness models were inferred based on the data from each subject, then more explanation is needed. In prior work, the use of these models to estimate fitness was justified by arguing that sequence variants common to many individuals are likely to be well-tolerated by the virus, while ones that are rare are likely to have high fitness costs. This justification is less clear for sequence variation within a single individual, where the viral population has had much less time to "explore" the sequence landscape. Nonetheless, there is precedent for this kind of analysis (see, e.g., Asti et al., PLoS Comput Biol 2016). If the authors took this approach, then this point should be discussed clearly and contrasted with the prior HIV and HCV studies.

We thank the reviewer for pointing out the weakness in our explanation and description of the fitness model. The model has been generated using publicly released viral sequences and this has been described in a previous publication by Hart et al. 2015. T/F virus from each of the subjects chronically infected with HCV in our cohort were given to the model by Hart et al. to estimate the initial viral fitness of the T/F variant. Subsequent time points of each subject containing the subvariants of the viral population were also estimated using the same model (each subtype). For each subject, these subvariant viral fitness values were divided by the fitness value of the initial T/F virus (hence relative fitness of the earliest time points with no mutations in the epitope regions were a value of 1.000). All other fitness values are therefore relative fitness to the T/F variant.

We have further clarified this point in the methods section “Estimating survival fitness of viral variant” to better describe how the data of the model was sourced (Lines 465-499).

To add to the reviewer’s point, we agree that sequence variants common to many individuals are likely to be well-tolerated by the virus and this event was observed in our findings as our data suggested that immune escape variants tended to revert to variants that were closer the global consensus strain. Our previous publications have indicated that T/F viruses during transmission were variants that were “fit” for transmission between hosts, especially in cases where the donor was a chronic progressor, a single T/F is often observed. Progression to immune escape and adaptation to chronic infection in the new host has an in-between process of genetic expansion via replication followed by a bottleneck event under immune pressure where overall fitness (overall survivability including replication and exploring immune escape pathways) can change. Under this assumption we questioned whether the observation reported in HIV studies (i.e. mutation landscapes that allow HIV adaptation to host) also happens in HCV infections. Furthermore, cohort used in this study is a rare cohort where patients were tracked from uninfected, to HCV RNA+, to seroconversion and finally either clearing the virus or progression to chronic infection. Thus, it is of importance to understand the difference between clearance and chronic progression.

Another important point for clarification is the definition of fitness. In the abstract, the authors note that multiple studies have shown that viral escape variants can have reduced fitness, "diminishing the survival of the viral strain within the host, and the capacity of the variant to survive future transmission events." It would be helpful to distinguish between this notion of fitness, which has sometimes been referred to as "intrinsic fitness," and a definition of fitness that describes the success of different viral strains within a particular individual, including the potential benefits of immune escape. In many cases, escape variants displace variants without escape mutations, showing that their ability to survive and replicate within a specific host is actually improved relative to variants without escape mutations. However, escape mutations may harm the virus's ability to replicate in other contexts. Given the major role that fitness plays in this paper, it would be helpful for readers to clearly discuss how fitness is defined and to distinguish between fitness within and between hosts (potentially also mentioning relevant concepts such as "transmission fitness," i.e., the relative ability of a particular variant to establish new infections).

Thank you for pointing out the weakness of our definition of fitness. We have now clarified this at multiple sections of the paper: In the abstract at lines 18-21 and in the introduction at lines 64-69.

These read:

Lines 18-21: “However, this generic definition can be further divided into two categories where intrinsic fitness describes the viral fitness without the influence of any immune pressure and effective fitness considers both intrinsic fitness with the influence of host immune pressure.”

Lines 64-69: “This generic definition of fitness can be further divided into intrinsic fitness (also referred to as replicative fitness), where the fitness of sequence composition of the variant is estimated without the influence of host immune pressure. On the other hand, effective fitness (from here on referred to as viral fitness) considers fundamental intrinsic fitness with host immune pressure acting as a selective force to direct mutational landscape (19)[REF], which subsequently influences future transmission events as it dictates which subvariants remain in the quasispecies.”

One concern about the analysis is in the test of Shannon entropy as a way to quantify the rate of escape. The authors describe computing the entropy at multiple time points preceding the time when escape mutations were observed to fix in a particular epitope. Which entropy values were used to compare with the escape rate? If just the time point directly preceding the fixation of escape mutations, could escape mutations have already been present in the population at that time, increasing the entropy and thus drawing an association with the rate of escape? It would also be helpful for readers to include a definition of entropy in the methods, in addition to a reference to prior work. For example, it is not clear what is being averaged when "average SE" is described.

We thank the reviewer to point out the ambiguity in describing average SE. This has been rectified by adding more information in the methods section (Lines 397 to 400):

“Briefly, SE was calculated using the frequency of occurrence of SNPs based on per codon position, this was further normalized by the length of the number of codons in the sequence which made up respective protein. An average SE value was calculated for each time point in each protein region for all subjects until the fixation event.”

To answer the reviewer’s question, we computed entropy at multiple time points preceding the observation in the escape mutation. The escape rate was calculated for the epitopes targeted by immune response. We compared the average SE based on change of each codon position and then normalised by protein length, where the region contained the epitope and the time it took to reach fixation. We observed that if the protein region had a higher rate of variation (i.e. higher average SE) then we also see a quicker emergence of an immune escape epitope. Since we took SE from the very first time point and all subsequent time points until fixation, we do not think that escape mutations already been present at the population would alter the findings of the association with rate of escape. Especially, these escape mutations were rarely observed at early time points. It is likely that due to host immune pressure that the escape variant could be observed, the SE therefore suggest the liberty of exploration in the mutation landscape. If the region was highly restrictive where any mutations would result in a failed variant, then we should observe relatively lower values of average SE. In other words, the higher variability that is allowed in the region, the greater the probability that it will find a solution to achieve immune escape.

**Reviewer #2 (Recommendations for the authors):**
In addition to the main points above, there are a few minor comments and suggestions about the presentation of the data.(1) It's not clear how, precisely, the model-based fitness has been calculated and normalized. It would be helpful for the authors to describe this explicitly. Especially in Figure 3, the plotted fitness values lie in dramatically different ranges, which should be explained (maybe this is just an error with the plot?).

We have now clarified how the model-based fitness has been calculated and normalized in the method section “Estimating survival fitness of viral variants” at line 465-472.

“The model used for estimating viral fitness has been previously described by Hart et al. (19). Briefly, the original approach used HCV subtype 1a sequences to generate the model for the NS5B protein region. To update the model for other regions (NS3 and NS2) as well as other HCV subtypes in this study, subtype 1b and subtype 3a sequences were extracted from the Los Almos National Laboratory HCV database. An intrinsic fitness model was first generated for each subtype for NS5B, NS3 and NS2 region of the HCV polyprotein. Then using, longitudinally sequenced data from patients chronically infected with HCV as well as clinically documented immune escape to describe high viral fitness variants, we generated estimates of the viral fitness for subjects chronically infected with HCV in our cohort.”

Our apologies, there was an error with the plot in Figure 3. This has now been resolved.

(2) In different plots, the authors show every pairwise comparison of ELISPOT values, population fitness, average SE, and rate of escape. It may be helpful to make one large matrix of plots that shows all of these pairwise comparisons at the same time. This could make it clear how all the variables are associated with one another. To be clear, this is a suggestion that the authors can consider at their discretion.

Thank you for the suggestion to create a matrix of plots for pairwise comparisons. While this approach could indeed clarify variable associations, implementing it is outside the scope of this project. We appreciate the idea and may consider it in future studies as we continue to expand on this work.